# Pine Wood Nematode's Migration and Defense Mechanism of Highly Resistant and Susceptible *Pinus massoniana*

Wenhua Li ⬤, Yifan Zuo, Lili Deng, Yini Xie, Kai Gao, Zhichun Zhou and Qinghua Liu *⬤

Research Institute of Subtropical Forestry, Chinese Academy of Forestry, Hangzhou 311400, China; lwh19980109@163.com (W.L.); zuoyifan993@163.com (Y.Z.); lili_deng1706@163.com (L.D.); xieyini0430@163.com (Y.X.); gaokai@caf.ac.cn (K.G.); zczhou_risf@163.com (Z.Z.)
* Correspondence: liuqh@caf.ac.cn

**Abstract:** Pine wilt disease has caused great economic loss and become an ecological threat since it was introduced into East Asia. In China, *Pinus massoniana* Lamb. is highly susceptible. The pathogenic of this disease is linked to the invasion of *P. massoniana* by the pine wood nematode (PWN, *Bursaphelenchus xylophilus*), leading to various physiological activities. However, the migration pathway of PWN and the defense mechanisms of *P. massoniana* tissue structure following invasion remain unclear. This knowledge is vital for understanding the pathogenesis of pine wood nematode disease. To address this issue, we analyzed the tissue structure damage, horizontal and vertical migration pathways, and histochemical reactions of *P. massoniana* after PWN inoculation. The results are as follows: susceptible *P. massoniana* exhibited more tissue structure damage compared to highly resistant *P. massoniana*. PWN reproduced and migrated by feeding on and damaging cells. In susceptible *P. massoniana*, PWN propagated and migrated throughout the entire plant. Highly resistant *P. massoniana* displayed limited horizontal and vertical migration of PWN, making it challenging for PWNs to move from cambium to xylem. After *P. massoniana* was damaged by PWNs, a protein cross-linking phenomenon appeared rapidly, with highly resistant *P. massoniana* exhibiting less protein cross-linking than the susceptible variety. Lignin synthesis is a crucial factor in the tissue defense of *P. massoniana*. Protein crosslinking provides time for lignin synthesis and is an vital component of tissue defense.

**Keywords:** pine wilt disease; tissue structure damage; protein crosslinking; lignification; defense mechanism





## 1. Introduction

Pine wilt disease, characterized by its rapid onset and high mortality [1], has inflicted significant damage on pine trees since its introduction in China. While some progress has been achieved in chemical and biological control, the disease continues to exhibit a propensity for spreading [2].

*Pinus massoniana* Lamb. is an evergreen tree belonging to the Pinaceae family, which is known for its quick-growth and drought-tolerant characteristics. It is a pioneer tree species of high-yield timber and tallow forest, which is widely distributed in China's subtropical regions [3–5]. The ninth forest resources inventory reveals that *P. massoniana* covers 8.04 million hectares. However, *P. massoniana* is susceptible to pine wilt disease, and dies within a short time after infection. It is an extremely dangerous epidemic, which seriously affects the balance of forest ecology and the sustainable development of forestry industry in China, causing huge economic losses and ecological threats [6].

At present, a number of highly resistant *P. massoniana* families and clones have been bred in China. After the invasion of PWNs, plant resistance is an important basis for resisting adverse environment [7–9]. The structural defense of *P. massoniana* extends from the outermost bark to the xylem. This defense is facilitated by the thickened cell walls

with lignification and embolization, which serve a hydrophobic function and act as a multifunctional barrier against external environmental factors [10]. PWNs are transmitted by *Monochamus alternatus*, which naturally invade pine trees through wounds in the tender branches when feeding on pine [11,12]. In this process, the tissue structure of pine from periderm to xylem serves as a "threshold" for PWN. The difference in tissue structure defense between highly resistant and susceptible *P. massoniana* determines the resistance of pine wood nematode to some extent. Rutherford et al. [13] believed that the rapid migration and reproduction of PWN in the host was an important factor promoting the development of the disease; Jin Gang [14] proposed that the incidence of *P. thunbergii* was directly related to the propagation and diffusion rate of nematodes in the tree; Mamiya [15] suggested that nematodes moved in pine within a short period of invasion, and with the propagation of PWN, pine physiology also changed. The fluorescence wheat germ lectin (F-WGA) with isothiocyanate can be bound to the epidermis of nematode, but not to plant tissue. Precise detection of PWN distribution and quantity is achievable, overcoming the limitations of the Baermann's funnel. However, the migration pathway of pine wood nematodes in the tree and the resistance defense mechanism of tissue structure of *P. massoniana* infected with pine wood nematodes remain unclear. In this study, we compared the tissue damage expansion and tissue response of different resistant *P. massoniana* inoculated with pine wood nematodes. Fluorescent staining clarified the migration pathway and reproduction process of PWNs, thus providing a theoretical basis for determining effective inhibition methods of pine wood nematodes.

## 2. Materials and Methods

### 2.1. Origin and Culture of Pine Wood Nematode

*Botrytis cineta* was inoculated on a potato glucose agar (PDA) medium and incubated in a mold incubator at 28 °C for 5 days. After the coloration, colonies grew into the upper lid of the petri dish, the highly pathogenic pine wood nematode "Guangde-3B" (mortality rates of susceptible *P. massoniana* were all 100% after inoculating with it at three sites. In the previous test, they were inoculated into the colonies of *B. cineta* and incubated at 28 °C for about 10 days, so that the colonies were completely eaten by PWNs. PWNs were isolated by Baermann's funnel [16]. The nematode culture medium was broken into a funnel with a padded filter, and ultrapure water was added to infiltrate the culture medium. After 6 h, the water stop clamp at the lower end of the funnel was opened to collect and separate part of the collected nematode, and then the water stop clamp was closed, and ultrapure water was added to infiltrate the culture medium. After 12 h, they were collected in a 10.0 mL centrifuge tube, rinsed with sterile water three times, and centrifuged at 300 RPM for 5 min to remove the supernatant. A suspension containing 50 pine wood nematodes per 1 μL was prepared by aggregating and mixing all the obtained centrifuge tubes.

### 2.2. Origin of Plant Material Nematode Inoculation, and Sampling

The experimental material was a 5-year old *P. massoniana* forest located in Linhai City, Zhejiang Province, at the forestry technology extension and nursery tourism service station. According to the resistance index of *P. massoniana* clones to pine wood nematode disease measured by Anhui Academy of Forestry in the early period, the clones of *P. massoniana* were divided into 1 to 4 grades, with grade 1 representing susceptibility and grade 4 indicating high resistance [17]. The five-year-old clones of *P. massoniana* Xiuning-5 and Huangshan-1 were from Fuyang, Zhejiang province in China, which Xiuning-5 features highly resistant and Huangshan-1 features susceptible characteristics. At about 30 days after inoculation of PWNs, the highly resistant clones were asymptomatic and susceptible *P. massoniana* wilted and withered (Figure 1). In the previous test, the morality rates of "Xiuning-5" were all 0% after inoculating with PWN at three sites; the mortality rates of "Huangshan-1" clones were all 100% after inoculation with PWN at three sites.

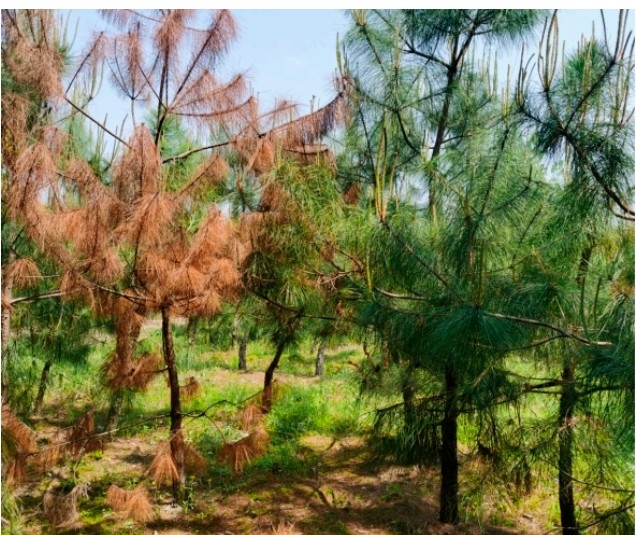

**Figure 1.** Resistant and susceptible symptoms of *P. massoniana*. Note: Left: Susceptible *P. massoniana*. Light: resistant *P. massoniana*.

On 28 July 2021, 10 highly resistant *P. massoniana* and 10 susceptible *P. massoniana* were selected on a sunny day with a temperature above 30 °C. Inoculation was performed using the peeling method, involving a 5 cm incision at the base of the annual shoot tooth marks in the wound to simulate the gnawing marks of the pine beetle were created by a file [18], and then 200 μL nematode suspension was evenly injected into the wound. After inoculation, the bark was covered over the wound and wrapped with cling film to ensure the survival of nematodes and to not be affected by the external environment.

The sampling location of *P. massoniana* included the site of inoculation, a site 1~3 cm above the inoculation point, the shoots that are 15~17 cm above the inoculation point, and the top of the shoots that are 30~35 cm above the inoculation point (Figure 2). The paraffin section prepared from the stem segment with highly resistant and susceptible *P. massoniana* before and after inoculation with pine wood nematodes at 1 d, 7 d, 15 d and 30 d. Three strains of the same clone were selected at each time point as biological replicates. These stem segments were collected and immediately put into the configured FAA fixing solution (70% tert-butanol: formaldehyde: propionic acid: glycerol = 17:1:1:1, *v/v/v/v*) for paraffin sections.

### 2.3. Paraffin Section Preparation and Staining

The fixed material was cut into slices with a thickness of 15 μm according to the paraffin section step [19]. Following the treatment, a Leica DM 4000 B microscope was used to observe and take photos. In order to observe the quantity and migration of pine wood nematodes, the paraffin sections after baking and dewaxing were cleaned three times with 0.01 M phosphate solution containing 0.05% Tween-20, soaked in 10 mM phosphate buffer containing 0.01% F-WGA for 1 h, and then cleaned three times with 0.01 M phosphate solution containing 0.05% Tween-20 [20]. A solution was prepared by adding 5.0 g of toluidine to 100.0 mL 30% ethanol. The solution was heated to 65 °C, stirred for 30 min, and then applied to dye the paraffin sections [21] for the purpose of observing the tissue and structural damage in *P. massoniana*; 100.0 mg Coomasil blue G-250 was dissolved in 50.0 mL 90% ethanol, then 85% phosphoric acid was added to 100.0 mL, and finally the paraffin sections were stained with distilled water to 1000.0 mL to observe the cross-linking of proteins in the tissue structure [22]; A solution was prepared by dissolving 1.0 g of phloroglucinol in 100.0 mL of 95% ethanol. The solution was used to stain for 2 min, after which 25% HCl was added for an additional 2 min to develop color and observe the accumulation of lignin in the tissue structure [23].

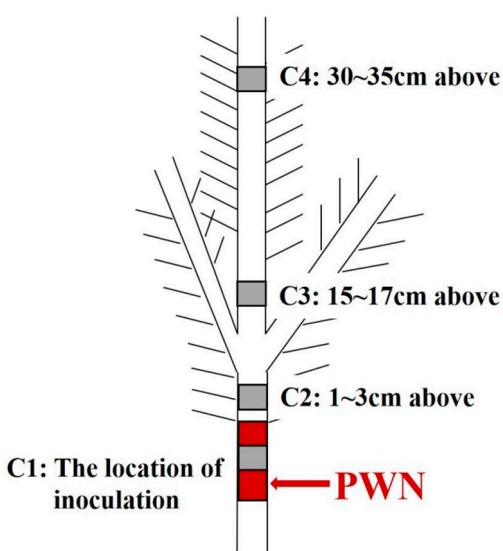

**Figure 2.** Inoculation and sampling site of *P. massoniana*. Note: The red part indicates the inoculating position. The gray part indicates the sampling position.

### 2.4. Tissue Damage Evaluation

The damaged cortical and xylem axial resin canals in the paraffin sections were observed and recorded microscopically. The resin canal area was measured using ImageJ. Tissue damage was assessed by calculating the ratio of the area of the axial resin canal in the damaged and undamaged cortex and xylem.

## 3. Results

### 3.1. The Tissue Structure Is Damaged and Dilated

The tissue structure of susceptible *P. massoniana* was damaged after inoculation with PWN, and the damage degree of susceptible *P. massoniana* was higher than that of highly resistant *P. massoniana* after inoculation with PWNs. The cortical cells and xylem cells of susceptible *P. massoniana* were damaged at 1 day after inoculation. The damage rate of *P. massoniana* with high resistance accounted for 1.15% (Table 1), and only the cells around the resin canal were damaged. At the same time, a few PWNs were observed in the canal (Figure 3A). After 7 and 15 days of inoculation with PWN, most epithelial cells in the cortex cells, xylem cells and resin canal in segments of susceptible *P. massoniana* 1~3 cm below the inoculation point were damaged by PWNs (Figures 3 and 4). Highly resistant *P. massoniana* showed damage in the cells around the resin canal and cortical cells at 7 days after inoculation. After 30 days of inoculation with PWN, the phloem cortex cells in segments of susceptible *P. massoniana* 30~35 cam below the inoculation point were damaged and a large number of cavities were generated (Figures 4 and 5A). In the contract, highly resistant *P. massoniana* exhibited a tissue structure damage ratio of 1.26%, with no expansion of the damaged area (Table 1).

**Table 1.** Tissue structure damage of *P. massoniana* inoculated with PWN with different resistance.

| Inoculation Time | Damage Ratio | | Coefficient of Variation | | *p* |
| --- | --- | --- | --- | --- | --- |
| | Resistant *P. massoniana* | Susceptible *P. massoniana* | Resistant *P. massoniana* | Susceptible *P. massoniana* | |
| 1 day | 1.15 | 3.83 | 23.07 | 41.47 | 0.07 |
| 7 day | 1.19 | 5.22 | 30.87 | 16.05 | 0.01 ** |
| 15 day | 1.25 | 6.74 | 82.42 | 42.99 | 0.02 * |
| 30 day | 1.26 | 9.36 | 46.24 | 35.20 | 0.02 * |

Note: "*" indicates significant at $p < 0.05$ level, "**" indicates significant at $p < 0.01$ level.

As for the degree of tissue structure damage, Table 1 showed that there was a significant difference in the tissue structure damage ratio between highly resistant and susceptible *P. massoniana* ($p < 0.05$) at 7, 15 and 30 days after inoculation with PWN, and the damage ratio of susceptible *P. massoniana* was higher than that of highly resistant *P. massoniana*.

### 3.2. Distribution of Pine Wood Nematode

#### 3.2.1. Horizontal Migration of Nematodes

In susceptible *P. massoniana*, 1 day after inoculated with PWN, PWNs were predominantly located in the cortical resin canal, and no PWN was found in other parts (Figure 3A). Notably, the tissue structure was damaged by PWNs (Figure 5B); 7 days after inoculation, the number of PWNs increased and PWNs distributed in the cortical resin canal, cortical cells, cambium, xylem cortical resin canal and xylem cells in large numbers (Figures 3 and 5C). After 15 days of inoculation, PWNs were distributed in all parts in susceptible *P. massoniana*, and began to appear in the pith (Figures 3D and 5D). After 30 days of inoculation with PWNs, the gap between cambium and xylem became larger, and there were a large number of PWN in cortical resin canal, cortical tissue, cambium, xylem cortical resin canal and pith.

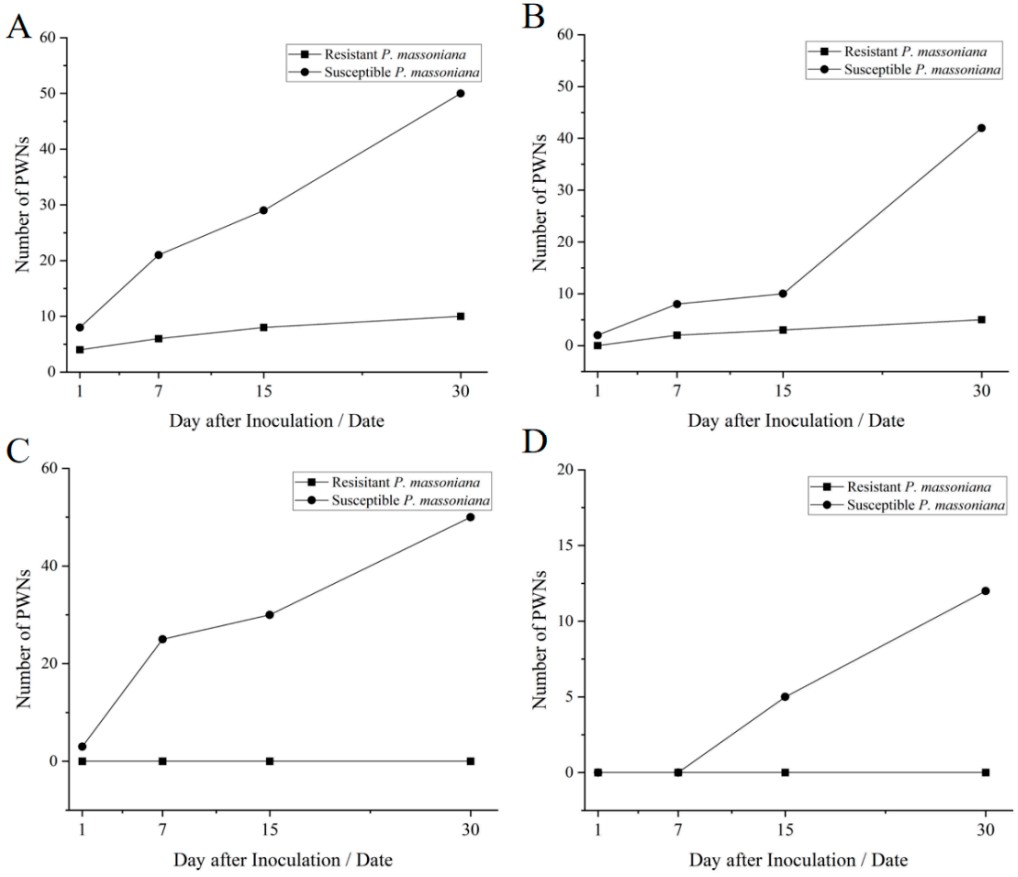

**Figure 3.** Number of pine wood nematodes with different tissue structure and different inoculation time of resistant and susceptible *P. massoniana*. Note: (**A**) The number of PWNs in cortical resin canal of *P. massoniana* after inoculation; (**B**) The number of PWNs in cambium of *P. massoniana* after inoculation; (**C**) The number of PWNs in xylem cortical resin canal of *P. massoniana* after inoculation; (**D**) The number of PWNs in the pith of *P. massoniana* after inoculation.

In highly resistant *P. massoniana*, 1 day after inoculation with PWN, a small number of PWNs were migrating in the cortical resin canal, but no PWN appeared in other parts (Figure 3A). Notably, the tissue structure was not damaged by PWNs (Figure 6); after 7 days of inoculation, the number of PWNs did not increase, and PWNs were distributed in

cortical cells and cortical resin canal (Figure 3C); after 15 days of inoculation, PWNs were able to migrate to cambium (Figure 3B); after 30 days of inoculation, the number of PWNs in all parts of *P. massoniana* with high resistance did not increase, and distribution positions did not spread (Figures 3 and 6).

### 3.2.2. Vertical Migration of Nematodes

In susceptible *P. massoniana*, 1 day after inoculation with PWN, PWNs congregated near the inoculation site (Figure 4A); after 7 days of inoculation, PWNs began to migrate and spread but were still limited to the inoculated branch (Figure 4B); after 15 days of inoculation, a few needle leaves turned yellow and withered, and the number of PWNs increased significantly: a few PWNs were able to migrate upward and downward for a short distance (Figure 4C); after 30 days of inoculation, the number of PWNs increased significantly, and a large number of PWNs were detected in all parts of the plant (Figure 4D).

In highly resistant *P. massoniana*, 1 day after inoculation with PWN, PWNs gathered near the inoculation site (Figure 4A); after 7 days of inoculation, PWNs began to spread but were confined to the vicinity of the inoculation site (Figure 4B); after 15 days of inoculation, there was no change in the number and distribution of PWNs (Figure 4C); after 30 days of inoculation, the number of PWNs remained unchanged, and they were confined to the area around the inoculation site (Figure 4D).

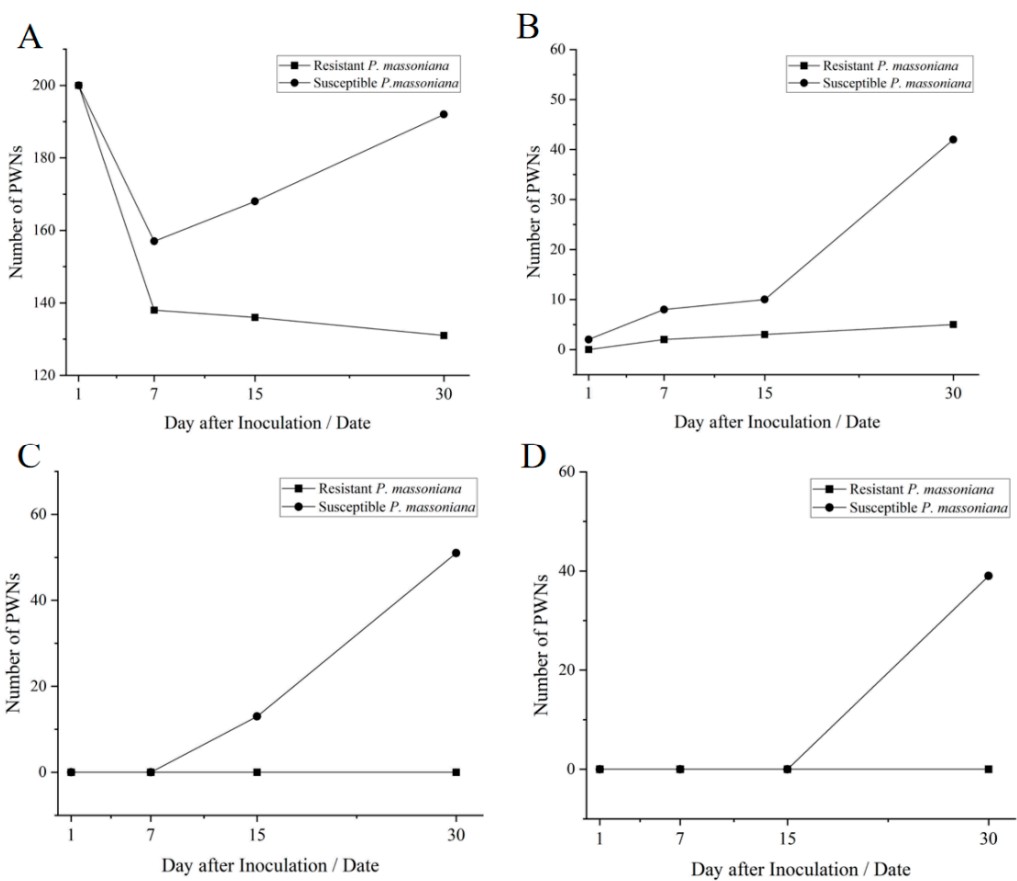

**Figure 4.** Number of pine wood nematodes with different sampling position and different inoculation time of resistant and susceptible *P. massoniana*. Note: (**A**) The number of PWNs in C1 site of *P. massoniana* after inoculation; (**B**) The number of PWNs in C2 site of *P. massoniana* after inoculation; (**C**) The number of PWNs in C3 site of *P. massoniana* after inoculation; (**D**) The number of PWNs in the C4 site of *P. massoniana* after inoculation.

### 3.3. Histochemical Reaction

The invasion of PWNs rapidly stimulated protein synthesis and the protein cross-linking phenomenon. After 1 day of inoculation, the protein cross-linking phenomenon was observed in cells around the damaged cambium and the intact cortical resin canal of susceptible *P. massoniana*. In contrast, highly resistant *P. massoniana* showed no protein cross-linking phenomenon except in the intact cortical resin tract (Table 2). After 7 days of inoculation, the protein cross-linking phenomenon was observed in cortical cells, damaged cambium and the pith of susceptible *P. massoniana* (Figure 5E). In highly resistant *P. massoniana*, only cortical resin canal and damaged cortical cells showed an obvious protein cross-linking phenomenon (Figure 6D). Significantly, the level of protein cross-linking phenomenon in susceptible *P. massoniana* was higher than that in highly resistant *P. massoniana* ($p < 0.05$).

**Table 2.** SDS staining of tissue structure in *P. massoniana* with different resistance after inoculation with pine wood nematodes.

| Inoculation Time | Unit Area Ratio/% | | CV/% | | $p$ |
|---|---|---|---|---|---|
| | Resistant *P. massoniana* | Susceptible *P. elliottii* | Resistant *P. massoniana* | Susceptible *P. elliottii* | |
| 1 day | 1.63 | 2.23 | 25.72 | 9.19 | 0.09 |
| 7 day | 2.54 | 4.07 | 12.99 | 11.71 | 0.06 |
| 15 day | 4.99 | 6.65 | 5.61 | 5.24 | 0.03 * |
| 30 day | 5.96 | 10.62 | 4.63 | 4.73 | 0.01 ** |

Note: "*" indicates significant at $p < 0.05$ level, "**" indicates significant at $p < 0.01$ level.

After inoculation of PWN, the accumulation of lignin occurred more slowly compared to the protein defense response. After 1 day of inoculation, there was no lignification observed in susceptible or highly resistant *P. massoniana* (Table 3). After 15 days of inoculation, there showed lignification in the cortical cells, the cell walls around the cortical cells, and the cells of the space between cambium and xylem of highly resistant *P. massoniana* (Figure 6E). In contrast, susceptible *P. massoniana* showed no lignification except in cortex cells (Figure 5F). Notably, highly resistant *P. massoniana* displayed significantly higher levels of lignification compared to the susceptible variety ($p < 0.05$).

**Table 3.** HCl-phloroglucinol staining of tissue structure in *P. massoniana* with different resistance after inoculation with pine wood nematodes.

| Inoculation Time | Unit Area Ratio/% | | CV/% | | $p$ |
|---|---|---|---|---|---|
| | Resistant *P.massoniana* | Susceptible *P.elliottii* | Resistant *P.massoniana* | Susceptible *P.elliottii* | |
| 1 day | 0.00 | 0.00 | - | - | - |
| 7 day | 5.14 | 3.90 | 13.17 | 17.29 | 0.09 |
| 15 day | 12.43 | 6.05 | 1.34 | 8.44 | 0.02 * |
| 30 day | 22.69 | 9.29 | 7.58 | 1.72 | 0.01 ** |

Note: "*" indicates significant at $p < 0.05$ level, "**" indicates significant at $p < 0.01$ level.

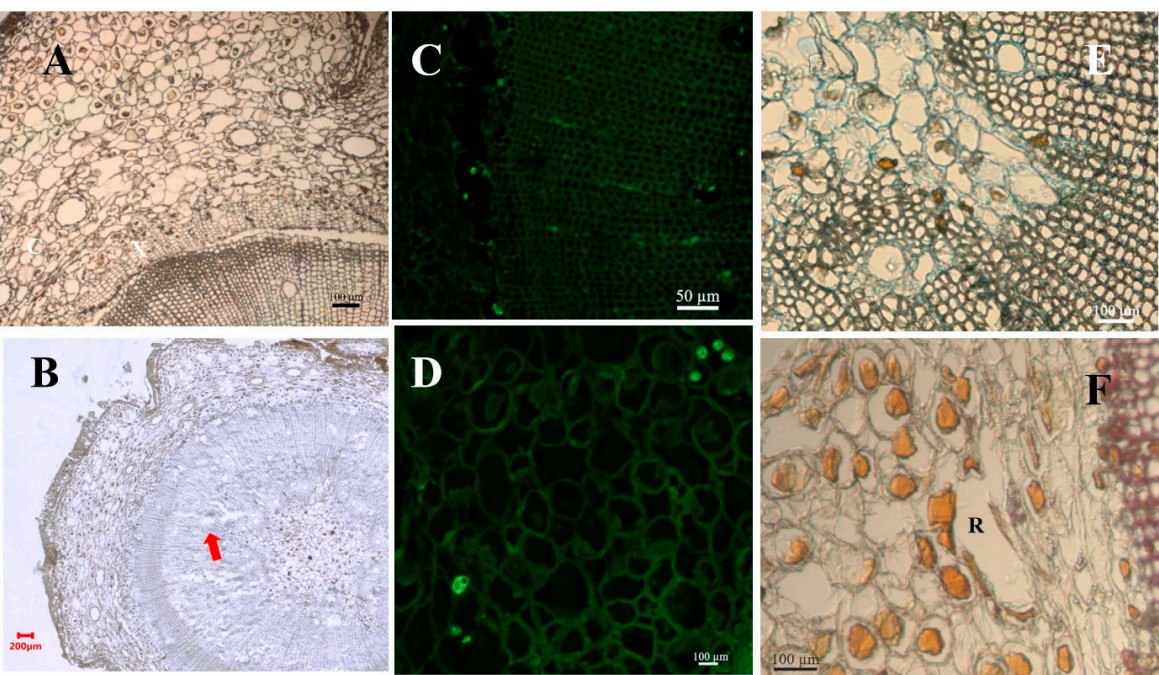

**Figure 5.** Tissue structure profile of susceptible *P. massoniana* after inoculation with pine wood nematodes. Note: Cross-sections of resistant *P. massoniana* stems inoculated with pine wood nematodes. (**A**) A partially non-stained section at 7 days after inoculation. (**B**) A partially non-stained section at 1 day after inoculation. The arrow pointed at the damage of the tissue structure. (**C**) Epifluorescent image of partially fluorescein isothiocyanate-conjugated wheat germ agglutinin (F-WGA)-stained section at 7 days after inoculation. Round green fluorescence indicated the presence of nematodes. (**D**) Epifluorescent image of partially fluorescein isothiocyanate-conjugated wheat germ agglutinin (F-WGA)-stained section at 30 days after inoculation. (**E**) SDS Coomassie blue stained section 7 days after inoculation. Blue color indicates accumulation of protein cross-linking. (**F**) Phloroglucinol-HCl stained section 15 days after inoculation. Red color indicates lignin accumulation. C indicated cambium. X indicated xylem. R indicated resin canal.

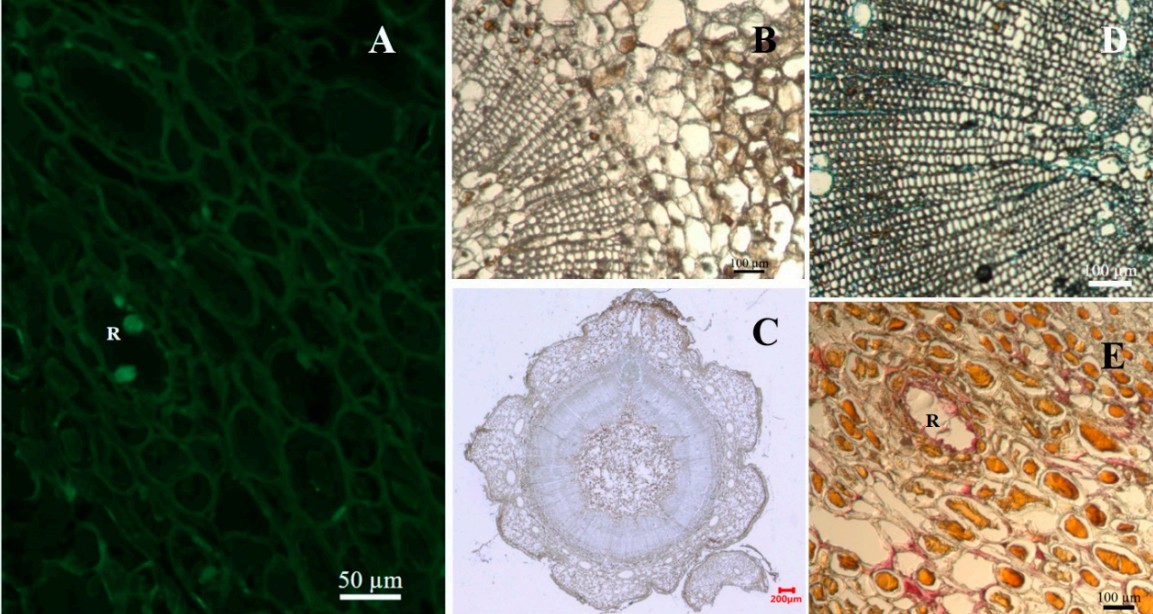

**Figure 6.** Tissue structure profile of *P. massoniana* with high resistance after inoculation with pine wood nematodes. Note: Cross-sections of resistant *P. massoniana* stems inoculated with pine wood

nematodes. (**A**) Epifluorescent image of fluorescein isothiocyanate-conjugated wheat germ agglutinin (F-WGA)-stained section at 30 days after inoculation. Round green fluorescence indicated the presence of nematodes. (**B**) A partially non-stained section at 1 day after inoculation. (**C**) A non-stained section at 7 days after inoculation. (**D**) SDS Coomassie blue stained section 7 days after inoculation. Blue color indicates accumulation of protein cross-linking. (**E**) Phloroglucinol-HCl stained section 15 days after inoculation. Red color indicates lignin accumulation. R indicated resin canal.

## 4. Discussion

In general, the damage expansion of the cortex, cambium and xylem parenchyma, including the axial resin canal, in resistant *P. massoniana* was slower than that in susceptible *P. massoniana* at the same time. At the later stage of infection, some PWNs were concentrated in the resin canal of resistant *P. massoniana*, while being distributed in the tissues of susceptible *P. massoniana*. This suggests that PWN primarily migrated and propagated in the resin canal. In line with previous studies, the migration rate of PWN in resistant species was slower than that in susceptible species after inoculation of PWN, and the degree of cell damage in resistant species was less than that in susceptible species within the same time [24–27]. The concentrated distribution of PWN in the space between the cambium and xylem of resistant *P. massoniana* indicated that PWN could not easily migrate horizontally from the cambium to xylem and pith. At 30 days after inoculation, PWN distribution was found in all tissues of susceptible *P. massoniana*, with the quantity significantly higher than in highly resistant *P. massoniana*. This difference may be attributed to the challenge of horizontal PWN migration in highly resistant *P. massoniana*. In addition, PWN diffusion sites in highly resistant *P. massoniana* were confined to inoculated branches, indicating an obstruction of vertical PWN migration. Son et al. [28] proposed that PWN invaded *P. thunbergii* and secreted a large amount of resin, thus interrupting the cortical resin tract and thus limiting the vertical migration of PWN.

Protein crosslinking refers to the process of forming covalent bonds between multiple skin chains within proteins or between protein molecules, which can significantly improve the functional properties of proteins. PWNs invaded *P. massoniana*, and protein cross-linking reaction occurred in the attacked cell wall within a short time [29]. In this study, at 7 days after inoculation, protein cross-linking of susceptible *P. massoniana* was significantly higher than that of highly resistant *P. massoniana* (Figures 4E and 5D) ($p < 0.05$). The protein crosslinking degree of highly resistant *P. massoniana* at 30 days after inoculation was similar to that of susceptible *P. massoniana* at 1 day after inoculation. These observations suggest that the expression of proteins related to cell wall strength is beneficial to limit migration and prevent cell destruction to a certain extent.

Lignin synthesis occurs continuously as phloem cells discolor and necrotize. Lignin synthesis and lignification a more time-consuming and require a more potent defense compared to protein-based defenses [30]. The pine resistance mechanism delays the rate of cell destruction, giving the tree enough time to accumulate lignin in its surrounding tissues. Phloem cells near the cambium of highly resistant *P. massoniana* exhibited extensive lignification at 15 days after inoculation, and PWNs were absent in the tissue structure of highly resistant *P. massoniana* 30 days after inoculation. Combined with the tissue damage of highly resistant *P. massoniana* inoculated with PWN, it can be inferred that highly resistant *P. massoniana* restricted PWN feeding behavior and inhibited its reproduction by generating lignin to strengthen the cell wall.

Kusumoto et al. [31] found that cell wall lignification effectively inhibited PWN migration and reproduction in resistant *P. thunbergii*. This study showed that highly resistant *P. massoniana* obstructed PWNs migration by impeding their progress through the cortex, cambium and xylem, which not only depended on protein defense, but also lignin defense. Xylem damage expansion eventually leads to tracheid embolism and the death of the tree [32]. If PWNs breach cambium defenses and regenerate, they can spread throughout the tree, potentially attacking living tissue on a large scale before lignin defenses come into play. Therefore, obstructing PWN migration from the cambium is crucial for

inhibiting PWN reproduction. Consequently, rapid prevention of tissue destruction after PWNs invasion is the pivotal factor in *P. massoniana* resistance against pine wilt disease.

## 5. Conclusions

By analyzing the damage of *P. massoniana* tissue structure, the horizontal and vertical migration of PWNs after inoculation, and the histochemical reaction of *P. massoniana* allowed us to speculate on the migration and feeding routes of pine wood nematodes after inoculation and defense mechanisms of *P. massoniana* tissue structure. The results showed that PWNs could not easily migrate horizontally from the cambium to xylem and pith. The PWNs diffusion site in highly resistant *P. massoniana* was confined to the inoculation site, which indicated that PWNs vertical migration was impeded. After the invasion of PWNs, protein cross-linking occurred rapidly and did not increase significantly over time. Moreover, the resistance mechanism of *P. massoniana* slows down the rate of cell destruction, providing sufficient time for the tree to accumulate lignin in the surrounding tissues. After the phloem cells became discolored and necrotic, lignin began to be synthesized. It can be inferred that highly resistant *P. massoniana* limits PWNs' feeding behavior by generating lignin to strengthen the cell wall, thus inhibiting their growth and reproduction. Therefore, preventing PWNs migration from cambium is particularly important for inhibiting PWNs reproduction. Our results provide an important scientific basis for the early control of pine wilt disease of *P. massoniana* in China.

**Author Contributions:** Q.L. and Z.Z. conceived of and designed the project. W.L., Y.Z. and L.D. collected the material. W.L., Y.Z. and Y.X. analysed the data and drew the diagram. W.L. and Q.L. wrote the manuscript. K.G., Z.Z. and Q.L. supervised the test. All authors have read and agreed to the published version of the manuscript.

**Funding:** This research was funded by the Zhejiang Science and Technology Program (2020C02007), the Forestry Science and Technology Innovation Special Project of Jiangxi Forestry Bureau (2021. No. 13), and the Zhejiang Science and Technology Major Program on Agricultural New Variety Breeding (2021C02070-5-2).

**Data Availability Statement:** The data that has been used is confidential.

**Acknowledgments:** We greatly thank Anhui Academy of Forestry for providing *Botrytis cineta* materials.

**Conflicts of Interest:** The authors declare no conflict of interests.

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
