# Peer review of "Pine Wood Nematode’s Migration and Defense Mechanism of Highly Resistant and Susceptible Pinus massoniana"

_forests, doi:10.3390/f14102108_

Round 1

Reviewer 1 Report

Manuscript contains interesting information on reaction of host plants to Bursaphelenchus xylophilus infestation. I think it could be published after some improvements, please see my suggestions:

 Introduction

PWN are not parasiting in M. alternatus, they are using the beetle as mean of transport only. B. xylophilus if phyto and mycophagous species of nematode.

l. 53 – incidence of black pine, please check

l 72 Bostonia cinerea – please correct

l 74 Baermann’s funnel

Resistance index – there are more approaches to concept of resistance in some resistant = no reproduction of pest and terms like level of tolerance are used.

Table 1 – significant correlation; I think you are not using correlation model, significant difference should be used

l 156 change the headline to “Horizontal migration of nematodes”

l. 170 were migrating

l 180 PWN turned yellow – branches turned yellow?

Figures – please check once more if really all figures are necessary for the paper, in some the description is not sufficient (eg. Figure 5, C what is the arrow pointing at?)

l 262 rosin

I am not English native speaker however I would strongly recommend the authors to let the manuscript check by professional English editing service; the manuscript contains some spelling errors and awkward formulations as I am able to judge.

Author Response

Manuscript contains interesting information on reaction of host plants to Bursaphelenchus xylophilus infestation. I think it could be published after some improvements, please see my suggestions:

Introduction

Point 1: PWN are not parasiting in M. alternatus, they are using the beetle as mean of transport only. B. xylophilus if phyto and mycophagous species of nematode.

Response 1: Thanks for your valuable comments. We have modified narrative in the manuscript. As follows: PWNs are transmitted by Monochamus alternatus, which naturally invade pine trees through wounds in the tender branches when feeding on pine. (line 49-52)

Point 2: 53 - incidence of black pine, please check

Response 2Thanks for your valuable comments. We have changed the “black pine” to “P. thunbergii” in the whole manuscript. (line 59-60)

Point 3: 72 Botrytis cineta-please correct

Response 3: We have changed the manuscript "Bostonia cinerea" to "Botrytis cineta". (line 75)

Point 4: 74 Baermann’s funnel

Response 4: We have changed the manuscript "Behrman funnel" to "Baermann’s funnel" in the whole manuscript. (line 65-66, 82)

Point 5: Resistance index - there are more approaches to concept of resistance in some resistant = no reproduction of pest and terms like level of tolerance are used.

Response 5: Thanks for your important suggestion. In this paper, the word “resistance index” is used to denote the level of tolerance of P. massoniana to pine wood nematodes.

Point 6: Table 1-significant correlation; I think you are not using correlation model, significant difference should be used

Response 6: We have changed the manuscript "“*” indicates that the correlation is significant at the p<0.05 level, “**” indicates that the correlation is significant at the p <0.01 level." to "“*” indicates significant at p<0.05 level, “**” indicates significant at p<0.01 level. (line 178)

Point 7: 156 change the headline to “Horizontal migration of nematodes”

Response 7: Thanks for your valuable suggestion. We have changed the manuscript "Pine wood nematode migrates horizontally" to "Horizontal migration of nematodes" (line 182). And we also changed the manuscript "Pine wood nematode migrates vertically" to "Vertical migration of nematodes". (line 199)

Point 8: 170 were migrating

Response 8: We have changed the manuscript "matriculis distributed" to "were migrating". (line 193)

Point 9: l 180 PWN turned yellow - branches turned yellow?

Response 9: Thanks for your suggestion. We have changed the manuscript "PWN" to "needle leaves". (line 203)

Point 10: Figures - please check once more if really all figures are necessary for the paper, in some the description is not sufficient (eg. Figure 5, C what is the arrow pointing at?)

Response 10: Thanks for your suggestion. In Figure 5C, the arrow is pointing at the damage of the tissue structure. We have canceled the unnecessary figures in the Figure 5 and Figure 6. (line 254, 268)

Point 11: 262 rosin

Response 11: We have changed the manuscript "rosin" to "resin". (line 299)

Comments on the Quality of English Language

Response: Thanks for your kind suggestion. We have repolished the article, please check.

Reviewer 2 Report

The authors analyzed the tissue structure damage, horizontal and vertical migration pathways of PWN, and histochemical reactions of highly resistant and susceptible P. massoniana after inoculation with pine wood nematodes.  The tissue structure damage of highly resistant P. massoniana was less than that of susceptible P. massoniana. The results showed that there was a significant difference in the tissue structure damage ratio between highly resistant and susceptible P. massoniana at 7, 15, and 30 days after inoculation with PWN. The damage expansion of cortex, cambium, and xylem parenchyma, including the axial resin tract, in resistant P. massoniana was slower than that in susceptible P. massoniana during the same time. The results showed that highly resistant P. massoniana inhibits the migration, growth, and reproduction of pine wood nematodes by producing lignin and strengthening cell walls. The scientific contribution of the work is to provide an important scientific basis for the control of pine wilt disease of P. massoniana.

Line 2 -3:  the title is not correct, migration refers to pine wood nematodes and the defense mechanism refers to P. massoniana

Line 13: the sentence should be written clearly (...migration pathways of PWN...)

Line 15: this sentence should be checked

Line 61: the sentence should be written more clearly (...tissue structure of P. massoniana  infected with pine wood nematodes...)

Line 69: the origin of the fungus Botrytis cinerea is missing

the word AGAR should be in lowercase letters

it would be good to explain why the fungus B.cinerea was used (As the pine wood nematode is mycophagous, it can propagate easily on cultures of Botrytis cinerea and other fungi.)

Line 71: the origin of the selected materials of highly pathogenic PWN should be indicated

Line 72: the name Bostonia cinerea is not correct

Lin 78:  the subtitle 2.2. should be adapted to the text (origin of plant material, nematode inoculation , and sampling)

It is not clear what the term DAI refers to. That needs to be corrected.

Lin 136: 1 DAI - did you mean in one day?

Lin 256: 30 DAI

Lin 268: 7 DAI

Lin 270: 30 DAI

Lin 271: 1DAI

Lin 279: 15 DAI

The cited literature in the text should be numbered.

References must be numbered in order of appearance in the text  and listed individually at the end of the manuscript.

In Reference, the term et al  (lines 323,325  etc) should be replaced with author names.

All authors should be listed in the References.

Author Response

The authors analyzed the tissue structure damage, horizontal and vertical migration pathways of PWN, and histochemical reactions of highly resistant and susceptible P. massoniana after inoculation with pine wood nematodes. The tissue structure damage of highly resistant P. massoniana was less than that of susceptible P. massoniana. The results showed that there was a significant difference in the tissue structure damage ratio between highly resistant and susceptible P. massoniana at 7, 15, and 30 days after inoculation with PWN. The damage expansion of cortex, cambium, and xylem parenchyma, including the axial resin tract, in resistant P. massoniana was slower than that in susceptible P. massoniana during the same time. The results showed that highly resistant P. massoniana inhibits the migration, growth, and reproduction of pine wood nematodes by producing lignin and strengthening cell walls. The scientific contribution of the work is to provide an important scientific basis for the control of pine wilt disease of P. massoniana.

Point 1: Line 2 -3: the title is not correct, migration refers to pine wood nematodes and the defense mechanism refers to P. massoniana

Response 1: Thanks for your valuable comments. We have changed the title “Migration and defense mechanism of highly resistant and susceptible pine wood nematodes in Pinus massoniana” to “Pine wood nematodes migration and defense mechanism of highly resistant and susceptible Pinus massoniana”. (line 2-4)

Point 2: Line 13: the sentence should be written clearly (...migration pathways of PWN...)

Response 2: Thanks for your valuable suggestions. We have modified narrative in the manuscript. As follows: the migration pathway of PWN and the defense mechanisms of P. massoniana tissue structure following invasion remain unclear. (line 13-15)

Point 3: Line 15: this sentence should be checked.

Response 3: Thanks for your valuable comment. We have modified narrative in the manuscript. As follows: Susceptible P. massoniana exhibited more tissue structure damage compared to highly resistant P. massoniana. (line 18-19)

Point 3: Line 61: the sentence should be written more clearly (...tissue structure of P. massoniana infected with pine wood nematodes...)

Response 3: Thanks for your comment. We have modified narrative in the manuscript. As follows: the migration pathway of pine wood nematodes in the tree and the resistance defense mechanism of tissue structure of P. massoniana infected with pine wood nematodes remain unclear. (line 66-68)

Point 4: Line 69: (1)the origin of the fungus Botrytis cinerea is missing; (2) the word AGAR should be in lowercase letters; it would be good to explain why the fungus B.cinerea was used (As the pine wood nematode is mycophagous, it can propagate easily on cultures of Botrytis cinerea and other fungi.)

Response 4: (1) Botrytis cineta was obtained from the Anhui Academy of Forestry in the early time. We breed regularly. And It is cultivated by adding bacteria from rotten apples to potato culture medium and then purifying and propagating it. (2) We have changed the manuscript "AGAR" to "agar". (line 75)

Point 5: Line 71: the origin of the selected materials of highly pathogenic PWN should be indicated.

Response 5: Thanks for your comment. We have modified narrative in the manuscript. As follows: “Guangde-3B” (mortality rates of susceptible P. massoniana were all 100% after inoculating with it at three sites in the previous test). These pine wood nematodes were isolated from the annual shoot P. massoniana in July 2021 and identified as "Guangde-3B".

Point 6: Line 72: the name Bostonia cinerea is not correct.

Response 6: We have changed the manuscript "Bostonia cinerea" to "Botrytis cineta". (line 75)

Point 7: Line 78: the subtitle 2.2. should be adapted to the text (origin of plant material, nematode inoculation , and sampling).

Response 7: Thanks for your valuable suggestions. We have modified narrative in the manuscript. As follows: Origin of plant material, nematode inoculation , and sampling. (line 91)

Point 8: It is not clear what the term DAI refers to. That needs to be corrected.

Lin 136: 1 DAI - did you mean in one day?

Lin 256: 30 DAI

Lin 268: 7 DAI

Lin 270: 30 DAI

Lin 271: 1DAI

Lin 279: 15 DAI

Response 8: Thanks for your valuable suggestions. We have changed manuscript "DAI" or “dai” to "day(s) after inoculation". (line 161, 167, 258, 259, 260, 262, 263, 264, 272, 273, 274, 275, 276, 292, 305, 308, 309, 317, 318)

Point 9: The cited literature in the text should be numbered. References must be numbered in order of appearance in the text and listed individually at the end of the manuscript.

Response 9: We have numbered the cited literature in the reference in the whole manuscript, please check.

Point 10: In Reference, the term et al (lines 323,325 etc) should be replaced with author names.

All authors should be listed in the References.

Response 10: We have listed all authors in the reference in the whole manuscript, please check.

Reviewer 3 Report

The manuscript enhances current knowledge about mitigation and resistance mechanism of PWN in Pinus massoniana. Overall the draft is of sound quality with appropriate design and interpretations. Some minor comments were outlined in the attached pdf. Please revise the manuscript according to the comments to make it suitable for publication. In addition to the comments in the pdf it would also be good to add more information about the isolate(s) of PWN used in the current study like their collection and pathogenicity etc.

Overall the English language is ok some small minor edits maybe required therefore it is recommended to once more proofread the entire manuscript for grammar and typo errors.

Author Response

The manuscript enhances current knowledge about mitigation and resistance mechanism of PWN in Pinus massoniana. Overall the draft is of sound quality with appropriate design and interpretations. Some minor comments were outlined in the attached pdf. Please revise the manuscript according to the comments to make it suitable for publication. In addition to the comments in the pdf it would also be good to add more information about the isolate(s) of PWN used in the current study like their collection and pathogenicity etc. 

Materials and methods

Origin and culture of pine wood nematode

Point 1: -Line 75: it would be relevant to add the information about when and from where they were collected.

Response 1: Botrytis cineta was obtained from the Anhui Academy of Forestry in the early time. We breed regularly. And It is cultivated by adding bacteria from rotten apples to potato culture medium and then purifying and propagating it.

Point 2: -Line 77: When and where was this selected from?

Response 2: These pine wood nematodes were isolated from the annual shoot P. massoniana in July 2021 and identified as "Guangde-3B".

Point 3: -Line 77: how the level of pathogenicity was deducted? 

Response 3: In the previous test, the mortality rates of susceptible P. massoniana were all 100% at three sites. Thus, we believed that “Guangde-3B” is a highly pathogenic pine wood nematode.

Point 4: -Line 82: Baermann, also please write briefly how it was done.

Response 4: Thanks for your valuable comments. We have changed the “Behrman” to “Baermann”. We have modified narrative in the manuscript. As follows: The nematode culture medium was broken into a funnel with a padded filter, and ultrapure water was added to infiltrate the culture medium. After 6h, the water stop clamp at the lower end of the funnel was opened to collect and separate part of the collected nematode, and then the water stop clamp was closed, and ultrapure water was added to infiltrate the culture medium. (line 82-86)

Point 5: -Line 88: RPM

Response 5: Thanks for your valuable comment. We have changed the manuscript "r·min-1" to "RPM" (line 88).

Point 6: -Line 121: Please mention whether the final data analyzed or used was it the average of the 3 replicates?

Response 6: The 3 replicates were used as the data for each clone. The final data analyzed was the 10 clones described above.

Results

Distribution of pine wood nematodes

Point 7: -Line 182: Horizontal migration of nematodes

Response 7: Thanks for your valuable suggestion. We have changed the manuscript "Pine wood nematode migrates horizontally" to "Horizontal migration of nematodes" (line 182).

Point 8: -Line 199: Vertical migration of nematodes

Response 8: Thanks for your valuable suggestion. We have changed the manuscript "Pine wood nematode migrates vertically" to "Vertical migration of nematodes" (line 199).
